SCUBE1 promotes pulmonary artery smooth muscle cell proliferation and migration in acute pulmonary embolism by modulating BMP7

Qu Xiaoya quxiaoya@mail.ustc.edu.cn
Huang Dongmei
Zhou Xiaomin
Ruan Wenwen
Department of Basic Medicine, Xiamen Medical College , Fujian , China
Zhan Cheng
Electronic publication date: 2024 Jan 19
Publication date: 2024
Volume: 12
Electronic Location ID: e16719
Received 2023 Oct 16; Accepted 2023 Dec 4
Copyright: ©2024 Qu et al.
Copyright year: 2024
Copyright holder: Qu et al.
License: This is an open access article distributed under the terms of the Creative Commons Attribution License, which permits unrestricted use, distribution, reproduction and adaptation in any medium and for any purpose provided that it is properly attributed. For attribution, the original author(s), title, publication source (PeerJ) and either DOI or URL of the article must be cited.
License URL: https://creativecommons.org/licenses/by/4.0/

Keywords: Acute pulmonary embolism, Pulmonary artery smooth muscle cell, SCUBE1, Proliferation, Migration

Funding: Science and Technology Project of Education Department of Fujian Province JAT200718 This study was supported by the Science and Technology Project of Education Department of Fujian Province (No. JAT200718). The funders had no role in study design, data collection and analysis, decision to publish, or preparation of the manuscript.

==============================
Objectives

After an episode of acute pulmonary embolism (APE), activated platelets have the ability to release various bioactive factors that can stimulate both proliferation and migration of pulmonary artery smooth muscle cells (PASMCs). SCUBE1 has been previously reported to engage in platelet-platelet interactions, potentially contributing to the activation of platelets in early onset thrombi. The purpose of this study was to examine the alterations in SCUBE1 expression in PASMCs after APE, as well as understand the mechanism behind these changes.

Methods

The platelet-rich plasma samples of both APE patients and healthy individuals were collected. A hyperproliferative model of PASMCs was established by using platelet-derived growth factor (PDGF) as a stimulator and various assays were used to investigate how SCUBE1-mediated BMP7 can regulate PDGF-induced PASMC proliferation and migration.

Results

Elevated level of SCUBE1 were observed in platelet-rich plasma from patients with APE and in PASMCs induced by PDGF. SCUBE1 interference ameliorated PDGF-driven cell proliferation and migration, and also downregulated PCNA expression. Additionally, mechanistic studies demonstrated that SCUBE1 could directly bind to bone morphogenetic protein 7 (BMP7) and enhance BMP7 expression, which completely abolished the impact of SCUBE1 silencing on proliferation and migration ability of PASMCs after PDGF treatment.

Conclusion

In the PDGF-induced proliferation of PASMCs, the expression of SCUBE1 and BMP7 was upregulated. Silencing of SCUBE1 impeded PDGF-induced proliferation and migration of PASMCs by restraining BMP7.

Introdution

Acute pulmonary embolism (APE) refers to a range of conditions or clinical syndromes in which different emboli occlude the pulmonary arterial system, resulting in morbidity as well as mortality (Lang et al., 2013; Raja et al., 2015). It is a common and severe life-threatening condition closely linked to myocardial infarction, stroke, and recognized as the three major cardiovascular diseases. In today’s era, people are facing an increasingly dangerous medical challenge due to the rapid onset of APE, various symptoms, high fatality rate, and frequent misdiagnosis and mistreatment (GBD 2019 Diseases and Injuries Collaborators, 2020; Van Galen et al., 2021). Despite improvements in diagnostic techniques, approximately 30% of patients still die without being diagnosed with APE (Nagamalesh et al., 2017). Currently, the existing drug treatment options for embolization are classified as thrombolytic therapy and anticoagulant therapy, based on the level of embolization. However, thrombolytic therapy is associated with prominent side effects and entails the potential risk of bleeding for patients. As a result, accurate diagnosis and search for novel therapeutic drugs and targets has become effective means to significantly improve the occurrence of thrombotic pulmonary arterial hypertension after APE.

With the advancement in biological research, researchers have discovered that PDGF, a major bioactive factor released by activated platelets after APE can exhibit a strong vasoconstrictor effect. In addition, studies progressively revealed that PDGF has the capacity to stimulate the growth of PASMCs and promote the conversion of fibroblasts into smooth muscle cells, which can lead to the remodeling of the pulmonary vasculature (Zhou et al., 2017; Xu et al., 2019). PASMCs are located in the tunica media of the pulmonary artery wall. Under pathological stimuli, PASMCs can effectively migrate from tunica media to intima, the interior of which can initiate the abnormal proliferation of PASMCs through a sequence of cellular signaling processes (Farkas & Kolb, 2011). Hence, underlying molecular mechanisms and signaling pathways that can contribute to abnormal proliferation and migration in PASMCs need to be deciphered.

Signal peptide cub epidermal growth factor domain containing protein 1 (SCUBE1), a glycoprotein found on the surface of activated platelets during early embryonic development can facilitate platelet adhesion by enhancing their interaction and supporting matrix adhesion (Tu et al., 2006). According to a previous study, an increase in SCUBE1 levels was observed in renal tumor tissues, suggesting that it can function as a potential biomarker in renal cell carcinoma (Karagüzel et al., 2017). Moreover, in aneurysmal subarachnoid hemorrhage (aSAH), the level of SCUBE1 was associated with greater severity and unfavorable outcomes of aSAH patients (Ding et al., 2016). Interestingly, some scholars have also reported that SCUBE1 levels were elevated in acute pulmonary embolism patients, but were extremely low in healthy people, thereby indicating that SCUBE1 could serve as a promising biomarker APE diagnosis (Wu et al., 2014). However, there are few studies related to SCUBE1 in APE, and its role in regulation of PASMC proliferation and migration needs further investigation to be fully understood.

Here, we aimed to decipher both role and regulatory network of SCUBE1 involved in the proliferation, migration of PASMCs, which could potentially understand the therapeutic potential of SCUBE1 in APE therapy.

Methods and Materials

Study subjects

The case group for this study consisted of APE patients who sought treatment at the Second Affiliated Hospital of Xiamen Medical College. Meanwhile, the control group was made up of individuals who had no prior allergies and were in good physical condition. The two groups were identical in terms of age and gender comparisons. All APE patients were diagnosed by computed tomography pulmonary arteriography and not administered any treatment before the diagnosis was confirmed. In addition, patients who have other forms of right ventricular dysfunction or other causes for SCUBE1 gain, such as acute coronary syndrome, acute myocardial infarction, acute ischemic cerebrovascular disease, peripheral artery disease and other ischemic conditions were excluded from this study. The venous blood (three mL) was collected from the APE and healthy control groups, respectively. The blood samples were then centrifuged, and the resulting upper plasma was preserved in a freezer at −80 °C for further use. All the samples obtained in this study were approved by the ethics committee of the Xiamen Medical College, complied with the ethical guidelines outlined in the Declaration of Helsinki and written informed consent was obtained.

Cell culture and treatment

Human PASMCs purchased from ATCC (VA, USA) were seeded into 6-well plates and cultured in a complete medium (DMEM/F12 medium + 10% FBS) at 37 °C and 5% CO2. For proliferation induction, different concentrations of PDGF (10, 20, or 40 ng/mL) were added to PASMCs. After growing the cells to 70%–80%, they were segregated into control group, PDGF group (PDGF (20 ng/mL) was added into PASMCs for 12 h of stimulation), PDGF + sh-NC group (24 h after PASMCs were transfected with sh-NC (5′-TTC TCC GAA CGT GTC ACG T-3′), PASMCs were stimulated after addition of 20 ng/mL PDGF for 12 h) and PDGF + SCUBE1 group (24 h after PASMCs were transfected with sh-SCUBE1(5′-GCT TTT CCT CCT CAT AAA T-3′), PASMCs were again stimulated with 20 ng/mL PDGF for 12 h). According to the Lipofectamine 2000 reagent instructions, sh-SCUBE1 and sh-NC were transfected into PASMCs and thereafter the transfected cells were cultured for 48 h.

RT-qPCR

MolPure® Cell/Tissue Total RNA Kit was used for total RNA extraction (Yi Sheng Biotechnology, shanghai, China). A prime script RT-PCR Kit (Takara, Dalian, China) was employed to reverse transcribe RNA into cDNA. The cDNA synthesized in the previous step was used as an amplification template for real-time PCR. The relevant reaction system was subsequently configured according to the SYBR Premix Ex TaqTM kit and then amplified using a fluorescence quantitative PCR machine. At the end of the reaction, the CT value was read and recorded, using a 2−ΔΔCT method to analyze the experimental results. The primer sequences used for RT-qPCR were as following: SCUBE1 forward, 5-AAC ACA CGG GTA CCG CCT CTT; SCUBE1 reverse, 5-GTA TTG TAG TGG TGT CCG GGA GA; β-actin forward 5′- CAT GTA CGT TGC TAT CCA GGC-3′; and β-actin reverse 5′-CTCCTTAATGTCACGCACGAT-3′.

Western blot assay

After transfection and treatment, PASMCs were added to RIPA lysate containing enzyme inhibitors and incubated on ice for 15 min. After the supernatant was collected by centrifugation, the concentration of each protein sample was measured after establishing a curve based on the absorbance of the standard. The protein lysates were diluted according to the protein concentration to ensure that identical amount of sample (20 ug) was loaded into each well, and then analyzed on a 10% SDS-PAGE gel. The separating glue was removed to facilitate wet membrane transfer. Thereafter, membrane was incubated in 5% skimmed milk powder blocking solution for 2 h, followed by the addition of the corresponding primary antibody (SCUBE1, ab105358, Abcam; PCNA, ab18197, Abcam; GAPDH, ab9485, Abcam; BMP7, ab84684, Abcam) for overnight incubation at 4 °C. The primary antibody was removed and the membrane was incubated with horseradish peroxidase-linked secondary antibody for 1 h. Finally, developer solution was added to the ChemiDoc Touch imaging system for subsequent exposure and imaging.

ELISA

A commercial ELISA assay (item No. CSB-E15005 h; Cusabio, Wuhan, China) was used to detect SCUBE-1 concentration in the plasma.

CCK-8 assay

PASMC suspension was added into individual wells with 4 × 103 cells seeded in each well. After 24 h, old medium was discarded and PASMCs were added to fresh medium with or without PDGF. Thereafter, experiments were terminated after stimulation at 24 h, 48 h, 72, and 96 h respectively. Each well was subsequently treated with 10 µL of CCK8 solution for a duration of 2 h and absorbance was measured at 450 nm using a microplate reader.

EDU assay

PASMCs were placed in 24 well plates and grown overnight. After that, a working solution of 10 µM EDU was added and the cells were incubated at 37 °C for 2 h. Thereafter, PASMCs were fixed through the addition of 4% paraformaldehyde for a duration of 15 min. After washing the PASMCs thrice with a 3% BSA solution, a permeabilization solution containing 0.3% Triton X-100 in PBS was added to each well and incubated for 15 min. Thereafter, 100 µl of click reaction solution was added and incubated in a wet box at the room temperature for 30 min under light. Finally, a 1:10 dilution of DAPI staining solution was added to each well and PASMCs were photographed under a fluorescence microscope.

Wound healing assay

After digestion and centrifugation, PASMCs were placed into six cm dishes at a suitable density. The growth of the cells was monitored by changing the medium every 2–3 days until they reached a confluence of 80–90%. A scratch was created by scratching the bottom of the dish with a 200 µL sterile pipette tip. The cells that were detached were rinsed with PBS. The culture dish was supplemented with medium, either with or without PDGF. Images were captured using an inverted microscope within 24 h after scratching. The measurement of relative distance of cell migration within the scratched area was conducted and the healing percentage was calculated.

Transwell assay

PASMC suspensions were made by digesting the cells to a final concentration of 5 × 105/mL. Next, 200 µL of the above cell suspension was added to the upper chamber of the well and 650 µL of the medium containing 10% FBS was added to the lower chamber. The chambers were then placed into a 5% CO2 incubator at 37 °C and incubated for 24 h. Subsequently, the remaining cells were gently wiped away using a disposable cotton swab. Thereafter, one mL of 4% paraformaldehyde was added, fixed for 20 min, and the chambers were stained with 0.5% crystal violet dye solution for 20 min. Finally, five different fields were selected for capturing images using an inverted microscope and the stained cells were counted.

Statistical analysis

The raw data obtained have been represented as mean ± standard (SD). The unpaired t-test feature of SPSS 20.0 (SPSS Inc., Chicago, IL, USA) was utilized to statistically compare the data between the two groups. When p value was <0.05, the difference was considered as statistically significant.

Results

SCUBE1 was highly expressed in patients with pulmonary embolism and in PDGF-induced PASMCs

To verify the expression status of SCUBE1 in the plasma APE patients, we employed ELISA and real time-PCR to determine the concentration of SCUBE1 and concentration of SCUBE1 and evaluate the expression levels of SCUBE1 mRNA respectively in the corresponding platelet-rich plasma. The findings indicated a significant increase in SCUBE1 mRNA levels in the platelet-rich plasma of APE patients when compared to healthy controls (Figs. 1A and 1B). Furthermore, PASMCs have been implicated to play a vital role in the development of APE. We stimulated the cells with different concentrations of PDGF (10, 20 or 40 ng/mL) to determine the expression of SCUBE1 in PASMCs. Western blotting data indicated that PDGF treatment caused an increase in SCUBE1 expression in PASMCs. The maximum impact was observed at a concentration of 20 ng/mL of PDGF (Fig. 1C).

Figure 1 SCUBE1 was overexpressed in the plasma of patients with APE and PASMCs after PDGF stimulation.

(A) ELISA was used to analyze the concentration of SCUBE1 in plasma (n = 50). (B) RT-qPCR was used to analyze the relative expression of SCUBE1 in platelet-rich plasma of APE (n = 50). (C) Western blotting was used to analyze relative expression of SCUBE1 in PASMCs after different concentration of PDGF (10, 20 or 40 ng/mL). *P < 0.05.

Interference of SCUBE1 restrained PDGF-induced proliferation of PASMCs

In order to determine the role of SCUBE1 in APE by regulating proliferation of PASMC, we utilized shRNA to disrupt the expression of SCUBE1 and then performed assays for proliferation and migration. The Western blotting results revealed that sh-SCUBE1 transfection effectively reduced SCUBE1 levels, while pcDNA-SCUBE1 transfection promoted SCUBE1 levels in PASMCs (Fig. 2A). CCK-8 assays showed that interference of SCUBE1 significantly suppressed growth of PASMCs, whereas overexpression of SCUBE1 had an opposite outcome (Fig. 2B). In addition, we also utilized the EDU assay to evaluate cell proliferation. The results demonstrated that cell proliferation was significantly increased in PDGF group compared to the control group. Conversely, cell proliferation was diminished in the PDGF + sh-SCUBE1 group compared to the PDGF + sh-NC group, while the cell proliferation of PDGF + pcDNA-SCUBE1 group was higher in comparison to PDGF + vector group (Fig. 2C). As an important protein regulating cell proliferation, the expression of PCNA was assessed by Western blotting. The findings revealed that PDGF had the ability to enhance PCNA expression. However, when SCUBE1 was inhibited, the level of PCNA decreased in the PDGF + sh-SCUBE1 group compared to the PDGF + sh-NC group. The results demonstrated that PDGF could promote the expression of PCNA, while the level of PCNA in the PDGF + sh-SCUBE1 group after inhibition of SCUBE1 was lower when compared to PDGF + sh-NC group. Conversely, when SCUBE1 was overexpressed, the PCNA level increased in the PDGF + pcDNA-SCUBE1 group compared to PDGF + vector group (Fig. 2D).

Figure 2 PDGF-induced proliferation of PASMCs was decreased by SCUBE1 silencing.

sh- SCUBE1, sh-NC, vector or pcDNA-SCUBE1 was transfected into PASMCs. (A) Western blotting detection of SCUBE1 level in PASMCs after treatment. (B) CCK-8 assay detection of PASMC proliferation after treatment. (C) EDU assay detection of PASMC proliferation after treatment. Scale bar = 20 µm. (D) Western blotting detection of PCNA level in PASMCs after treatment. *P < 0.05.

Interference of SCUBE1 restrained migration of PASMCs

Following the transfection of SCUBE1 shRNA into PASMCs, the migration ability of these cells was measured using the Transwell assay and wound-healing assay in each group. By conducting Transwell assay, it was determined that SCUBE1 silencing lead resulted in decrease in the migration ability of PASMCs, and caused a decline in the number of cells passing through the chamber, however, SCUBE1 overexpression lead to opposite results (Fig. 3A). In addition, the findings from the wound-healing assay provided strong evidence that inhibiting SCUBE1 significantly hindered the migration capability of PASMCs, whereas overexpression of SCUBE1 showed the opposite effect (Fig. 3B).

Figure 3 PDGF-induced migration of PASMCs was decreased by SCUBE1 silencing.

sh- SCUBE1, sh-NC, vector or pcDNA-SCUBE1 was transfected into PASMCs. (A) Transwell migration assay detection of PASMCs migration. Scale bar =100 µm. (B) wound-healing assay detection of PASMCs migration. Scale bar =200 µm. *P < 0.05.

BMP7 could be targeted by SCUBE1

We also investigated the biological mechanism of SCUBE1 involved in APE progression. BMP7 has been previously identified as a potential downstream target of SCUBE1. Western blotting assays demonstrated that treatment with PDGF increased the levels of BMP7 in PASMCs in comparison to the control group (Fig. 4A). By conducting a co-IP assay analysis, we discovered that BMP7 was detected in complexes upon precipitation with antibody against SCUBE1, whereas SCUBE1 existed in complexes upon precipitation with antibody against BMP7 in comparison with control IgG (Figs. 4B and 4C). Furthermore, the results of western blotting revealed that elevated levels of SCUBE1 led to an upregulation of BMP7 expression in PASMCs (Fig. 4D).

Figure 4 SCUBE1 bound with BMP7.

(A) Western blotting detection of BMP7 level in PASMCs after PDGF treatment. (B and C) CoIP validation of the relationship of SCUBE1 and BMP7. (D) Western blotting detection of BMP7 level in PASMCs after SCUBE1 overexpression or knockdown. *P < 0.05.

SCUBE1 contributed to PASMC proliferation and migration by regulating BMP7 expression

To examine the role of BMP7 in the biological actions of SCUBE1 in PASMCs, sh-SCUBE1 was transfected alone or in combination with BMP7 in PASMCs. Western blot data revealed that inhibition of SCUBE1 significantly inhibited BMP7 expression, which was restored upon transfection with BMP7 (Fig. 5A). Moreover, by utilizing CCK-8 and EDU assays, we noted that introduction of BMP7 reversed the impact of SCUBE1 silencing on proliferation ability of PASMCs (Figs. 5B and 5C). Interestingly, the silencing of SCUBE1 resulted in a noticeable reduction in PCNA expression in PASMCs, whereas BMP7 upregulation could reverse these changes (Fig. 5D). In addition, the migration ability of SCUBE1-depleted PASMCs was increased, but was abrogated by BMP7 upregulation (Figs. 5E and 5F).

Figure 5 SCUBE1 interference accelerated proliferation and migration in PDGF-induced PASMCs by reducing BMP7.

(A) Western blotting detection of BMP7 level in PASMCs. (B) CCK-8 and EDU detection of PASMCs proliferation. Scale bar = 20 µm. (D) Western blotting detection of PCAN level in PASMCs. (E and F) Transwell and Wound-healing assay detection of PASMCs migration. Scale bar = 100 µm; Scale bar = 200 µm. *P < 0.05.

Discussion

APE is a common complex cardiovascular disease, which primarily arises due to numerous pathogenic factors, with pulmonary vascular remodeling being its main feature. In recent years, some drugs have been proven effective in improving clinical outcomes. However, the overall prognosis of this disease remains poor, and there is limited understanding of its underlying pathogenesis. One of the important mechanisms involved in the local vascular remodeling of pulmonary arteries, following APE, is the hyperproliferation and suppression of apoptosis in PASMCs. Hence, currently, the focus of APE therapy research is on finding effective methods to inhibit proliferation and enhance apoptosis of PASMCs.

An escalating number of researchers have directed their attention towards the aberrant proliferation of PASMCs. It has been demonstrated that miR-106b-5p expression was significantly downregulated in PDGF-induced PASMCs as well as in APE mouse models and pharmacologically targeting NOR1 inhibited both cell proliferation and migration (Chen et al., 2020). Silencing miR-34a-3p led to an increase in DUSP1 level, which further facilitated the development of APE. This was manifested by the acceleration of mPAP elevation and thickening of the pulmonary artery wall in vivo, as well as the promotion of PASMC proliferation and migration in vitro (Li et al., 2022). Interestingly, a recent study has reported that let-7b-5p can affect proliferation and migration of PASMCs. Its ability to inhibit the expression of IGF-1 can effectively restrict the proliferation and migration of PASMCs (Zhang et al., 2022). Moreover, Janus kinase 3 inhibition by JANEX-1 led to attenuation of the proliferation of PASMCs induced by PDGF through modulating the STAT3/VEGF/FAK signaling pathway in APE (Pan et al., 2020). In addition, Toprak et al. (2022) demonstrated that that COVID-19 patients who experienced thrombotic complications had significantly elevated levels of SCUBE1 in their plasma. Furthermore, Wu et al. (2014) reported that deficiency of SCUBE1 in mouse plasma decreased the occurrence of arterial thrombosis and prevented fatal thromboembolism caused by collagen-epinephrine treatment. Our data demonstrated that increase in SCUBE1 levels observed in patient platelet-rich plasma could be used for the early diagnosis of APE. In addition, findings of cellular functional studies revealed that SCUBE1 interference was able to significantly attenuate the promoting effect of PDGF on both proliferation and migration of PASMCs.

In recent studies, SCUBE1 has been strongly linked to various cardiovascular diseases like acute ischemic stroke, APE, deep vein thrombosis, and acute coronary syndrome (Wu et al., 2014; Dai et al., 2008; Turkmen et al., 2015). Importantly, SCUBE1 can function as an important adhesion protein by using its EGF-like domain to form cross homophilic bridges during thrombus formation, consequently causing development of acute thromboembolic disease (Tu et al., 2006; Wu et al., 2014; Tu et al., 2008). In addition, following renal I/R injury, an increase in SCUBE1 expression was found in the peritubular capillaries, which effectively stimulated the proliferation of epithelial cells through BMP7 signaling (Zhuang et al., 2010; Liao et al., 2019). Interestingly, prior studies have indicated that bone morphogenetic proteins (BMPs) can modulate angiogenesis by influencing growth, differentiation, and turnover of vascular cell populations (Morrell, 2006; Moser & Patterson, 2005). Furthermore, evidences have suggested that dysregulation of bone forming protein signaling was associated with abnormal proliferation and migration of PASMCs. For example, BMP2 was found to be selectively upregulated in pulmonary arteries exposed to hypoxia, and it could effectively suppress PASMC proliferation and promote apoptosis (Takahashi et al., 2006). BMP4 can also contribute to the development of chronic hypoxic pulmonary hypertension by stimulating the proliferation, migration, and vascular remodeling of PASMCs (Zhang et al., 2013; Anderson et al., 2010). Importantly, BMP7 was observed to be increased in PASMCs following monocrotaline pyrrole (MCTP) stimulation, leading to enhanced cell proliferation and elevated PCNA expression. This was achieved through the inhibition of BMPR2 and the upregulation of ActIIα levels (Sun et al., 2019). In this study, we found evidence suggesting that SCUBE1 can directly affect the expression of BMP7, as silencing of SCUBE1 resulted in a decrease in BMP7 levels. Overall, knocking down SCUBE1 inhibited the impact of PDGF on proliferation and migration of PASMCs through silencing of BMP7.

Conclusions

PDGF treatment resulted in increased expression of SCUBE1, consequently stimulating the expression of BMP7. The positive regulation of BMP7 by SCUBE1 may enhance PASMC proliferation induced by PDGF, thereby promoting further progression of APE. The findings can provide novel targets and molecular markers, which can be used for the prevention and treatment of APE. However, it is worth noting that we have conducted preliminary investigations into the possible involvement of SCUBE1 in APE only under in vitro settings, which still needs to be confirmed in vivo.

Supplemental Information

Data S1 Raw Data

Click here for additional data file.

Supplemental Information 2 Western blot images

Click here for additional data file.

Supplemental Information 3 MIQE checklist

Click here for additional data file.

I am incredibly grateful for the help provided by my lab members while compiling this thesis. I would also like to thank my colleagues, Huaxin Liu and Chun Wu, for their invaluable assistance and guidance in my academic pursuits.

Additional Information and Declarations

Competing Interests

Author Contributions

Human Ethics

Data Availability

The authors declare there are no competing interests.

Xiaoya Qu conceived and designed the experiments, performed the experiments, analyzed the data, prepared figures and/or tables, authored or reviewed drafts of the article, and approved the final draft.

Dongmei Huang conceived and designed the experiments, analyzed the data, authored or reviewed drafts of the article, and approved the final draft.

Xiaomin Zhou performed the experiments, analyzed the data, prepared figures and/or tables, and approved the final draft.

Wenwen Ruan conceived and designed the experiments, performed the experiments, prepared figures and/or tables, authored or reviewed drafts of the article, and approved the final draft.

The following information was supplied relating to ethical approvals (i.e., approving body and any reference numbers):

All samples obtained in this study were approved by the ethics committee of the Xiamen Medical College and abided by the ethical guidelines of the Declaration of Helsinki.

The following information was supplied regarding data availability:

The raw data is available in the Supplementary Files.

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
