# Peer review of "SCUBE1 promotes pulmonary artery smooth muscle cell proliferation and migration in acute pulmonary embolism by modulating BMP7"

_PeerJ, doi:10.7717/peerj.16719_

## Round 0.1 · original submission · Major Revisions

Please revise the manuscript as the reviewers suggested.

**Language Note:** PeerJ staff have identified that the English language needs to be improved. When you prepare your next revision, please either (i) have a colleague who is proficient in English and familiar with the subject matter review your manuscript, or (ii) contact a professional editing service to review your manuscript. PeerJ can provide language editing services - you can contact us at [email protected] for pricing (be sure to provide your manuscript number and title). – PeerJ Staff

Reviewer 1 ·

Basic reporting

The paper by Qu et al. is an interesting study, which investigated the role of SCUBE1, a glycoprotein, on pulmonary artery smooth muscle cell. Actually, they are trying to report that mechanism of the effect of SCUBE1 for the first time. As a whole, their experiments are well-designed This is an interesting study. Here, I just I give some suggestions to improve the manuscript.
Minor:
1. In the conclusion of abstract, the sentence seems too long and I suggest modifying it so it can be easier connected to the results.
2. Line 55-60: I suggest splitting the sentence and using an alternative instead. For example: “With the development of biological research, researchers have found that platelet-derived growth factor (PDGF), a major bioactive factor released by activated platelets after APE, has a strong vasoconstrictor effect. Studies progressively revealed that PDGF is also able to promote the proliferation of pulmonary artery smooth muscle cells (PASMCs) and ……”.
3. Line 60-64: This sentence is too long so I suggest rewriting it or split on 2.
4. Line 81-82: I suggest that this sentence can be replace by: …. PASMCs and may provide a ……
5. Line 85: Acute pulmonary embolism instead of APE; “The” should be delete.

Experimental design

1. Please provide the sequence of sh-SCUBE1 and sh-NC for the method section.
2. Please add the corresponding scale and the zoom scale of the picture on the figure legends or picture.
3. Why do the authors describe cell culture at different time points (24, 48, 72h) in 2.5 cck-8, but 96 h appears in Figures 2 and 5?

Validity of the findings

1. Similar studies were found in PubMed. Authors shall cite part of them and discuss what is new and different from this article.
2. Prepare a paragraph of limitations in Section discussion.

Additional comments

No.

·

Basic reporting

The manuscript is a study of the mechanism of SCUBE1 in the progression of APE. The experimental design is reasonable, and the results are reliable. However, I have the following suggestions:

1. The primary focus of this study is the impact of SCUBE1 on PASMC proliferation and migration, which are fundamentally the main mechanisms for the progression of APE to pulmonary arterial hypertension. Therefore, I believe the value of this research lies in "whether it can improve the occurrence of thrombotic pulmonary arterial hypertension after APE" rather than what was stated in the last sentence of the first paragraph of the introduction, "to reduce the case fatality of acute pulmonary embolism."
2. Please provide the primer sequences used for RT-qPCR.
3. There is an extra word "in" in the title of Figure 1. The English grammar in the titles and captions of all figures needs to be revised.
4. It is not clear from Figure 1B why the dose of 20ng/ml was chosen for subsequent experiments. Multiple pairwise comparisons are needed.
5. The fluorescence intensity characteristics in the EdU images of the PDGF+sh-NC group in Figure 2 and the PDGF+sh-SCUBE1+BMP7 group in Figure 5 do not seem very distinct.
6. The layout of Figure 5 appears to be somewhat disorganized.
7. Do the last two sentences of the final paragraph in the discussion section align with the research results?
8. Similar to point 1, in the conclusion, it is mentioned, "SCUBE1 might contribute to the PDGF-induced proliferation of PASMCs by positively regulating BMP7, which in turn promoted the occurrence and development of APE." However, I believe this study can only speculate that SCUBE1 plays a role in the further progression of APE, rather than in the occurrence of APE.

There are some minor revisions; please refer to the reviewed manuscript for details.

Experimental design

No comment.

Validity of the findings

No comment.

Additional comments

No comment.

Reviewer 3 ·

Basic reporting

In this paper, the authors look at how SCUBE1 (signal peptide-CUB-EGF-like repeat-containing protein 1) affects PASMCs (pulmonary artery smooth muscle cells) in vitro in response to PDGF (platelet-derived growth factor). In the context of PDGF-induced PASMC proliferation and migration, the effect of SCUBE1 on BMP7 expression was also investigated. The authors basically concluded that SCUBE1 is involved in PDGF-stimulated proliferation and migration of PASMCs via positive BMP7 overexpression. There are numerous concerns / flaws with this manuscript that preclude it from being published in a scientific journal.

1. This study did not report the underlying mechanism of SCUBE1-mediated upregulation of BMP7 in PDGF-treated PASMCs.
2. How could the mRNA expression level of SCUBE1 be measured in the plasma samples? The secreted SCUBE1 protein can be measured in the circulation but not the mRNA level.
3. BMP7-triggered pSMAD1/5/8 should be determined in PDGF-treated PASMCs.
4. So many typos and mis-labeled figure in the text. For example, line 216 and abstract, PCNA not PCAN; line 176, Figure 1B not 1A; line 202, Figure 4A not 3A; line 205, Figure 4B and C not 3B and C; line 207, Figure 4D not 3D; line 215, Figure 5B and C not 4B and C etc.
5. The authors should use 2 different ShRNA clones for knockdown experiments.
6. Gain-of-function by SCUBE1 overexpression is required to verify the function of SCUBE1 on PDGF-induced PASMC proliferation and migration.
7. Fluorescent cell images in Figure 2C and Figure 5 needs to improve.
8. English editing service is required.
9. Title of the manuscript is inconsistent with the conclusion.
10. The authors should specify the sources of commercial antibodies (catalog numbers) for SCUBE1, BMP7, PCNA… etc

Experimental design

Please see above Basic reporting.

Validity of the findings

Please see above Basic reporting.

---

## Round 0.2 · Minor Revisions

Please revise the manuscript as the reviewers suggested.

Reviewer 1 ·

Basic reporting

After modification, the structure of the article became clear and clear, using professional English throughout.
References provide sufficient on-site background.

Experimental design

After modification, the research question is clearly defined, relevant, and meaningful. It explains how research can fill identified knowledge gaps.
The described method has sufficient details and information for replication.

Validity of the findings

All underlying data have been provided; they are robust, statistically sound, & controlled.
Conclusions are well stated, linked to original research question & limited to supporting results.

Additional comments

I don't have any additional review comments, I believe the article meets the publication standards.

·

Basic reporting

The concerns and suggestions I had have been largely addressed in the revised manuscript. However, in light of the comments from Reviewer 3, I also believe that the authors should provide additional reasonable explanations to demonstrate why SCUBE1 mRNA can be detected in the plasma.

Experimental design

Please see above.

Validity of the findings

Please see above.

Additional comments

Please see above.

---

## Round 0.3 · accepted · Accept

This manuscript can be accepted now.